# Neurological Consequences of Pulmonary Emboli in COVID-19 Patients: A Study of Incidence and Outcomes in the Kingdom of Saudi Arabia

**DOI:** 10.3390/brainsci13020343

**Published:** 2023-02-17

**Authors:** Ebtisam Bakhsh, Mostafa Shaban, Mohammad Abdullah Alzoum, Areej M. AlNassir, Aliah A. Bin Hamad, Munira S. Alqahtani, Leenah Ayman F. AlAyoubi, Raghad Mohammed Alamri, Nasser F. Alamri

**Affiliations:** 1Clinical Sciences Department, College of Medicine, Princess Nourah bint Abdulrahman University, Riyadh 11671, Saudi Arabia; 2Community Health Nursing Department, College of Nursing, Jouf University, Sakaka 72388, Saudi Arabia; 3King Abdullah bin Abdulaziz University Hospital, Riyadh 11552, Saudi Arabia; 4Cardiothoracic Imaging Department, King Fahad Medical City, Riyadh 11525, Saudi Arabia

**Keywords:** COVID-19, pulmonary embolism, neurological complications

## Abstract

Pulmonary embolism (PE) is a significant consequence that is becoming more common in COVID-19 patients. The current study sought to determine the prevalence and risk factors for PE in a study population of COVID-19 patients, as well as the relationship between PE and neurological sequelae. The research also sought to analyze the consistency of neurological examination and imaging techniques in detecting neurological problems. The research comprised a total of 63 individuals with COVID-19. The incidence of PE in the study group was 9.5% for smokers, 23.8% for obese patients, 33.3% for hypertensive patients, and 19% for diabetic patients, according to the findings. After adjusting for possible confounders such as age, gender, BMI, smoking, hypertension, and diabetes, a logistic regression analysis indicated that the probabilities of having neurological complications were 3.5 times greater in individuals who had PE. In conclusion, the present study highlights the high incidence of PE among patients with COVID-19 and the association between PE and neurological complications. The study also emphasizes the importance of a thorough neurological examination and imaging studies in the detection of neurological complications in patients with PE.

## 1. Introduction

COVID-19, which is caused by the SARS-CoV-2 virus, has had a major influence on world health. COVID-19 can induce a range of additional problems, including blood clots, in addition to respiratory difficulties [1]. COVID-19 has been linked to pulmonary emboli, or blood clots in the lungs, which can have significant implications, including death. However, little is known regarding the prevalence and consequences of pulmonary emboli in COVID-19 patients in Saudi Arabia [2]. Furthermore, the neurological implications of pulmonary emboli in COVID-19 patients have received less attention [3].

The SARS-CoV-2 virus causes pulmonary emboli (PE), a severe consequence of COVID-19. PE is a disorder in which blood clots develop in the deep veins of the legs, arms, or other regions of the body and travel to the lungs, where they impede blood flow [4]. PE can cause shortness of breath, chest discomfort, and even death in extreme cases [5].

PE has been observed to be more common in COVID-19 patients than in the general population. This is most likely related to COVID-19 patients’ hypercoagulable status, which raises the danger of blood clots [6]. Furthermore, COVID-19 individuals may be more vulnerable to PE owing to extended immobility, such as during hospitalization or bedrest [7].

The frequency of PE in COVID-19 patients is greater than in the general population, increasing the risk of neurological sequelae [8]. PE has been linked to an increased risk of stroke among COVID-19 patients [9], according to research. This association’s mechanism is unknown; however, it is considered to be connected to the hypercoagulable condition that can arise in COVID-19 patients, which raises the risk of blood clots [6,10].

Recent studies have also reported that the COVID-19 virus can lead to neurological complications, including encephalitis, peripheral neuropathy, and stroke. It is believed that the inflammatory response caused by the virus may play a key role in the development of these neurological consequences [11,12].

Inflammation is a crucial component of the body’s immune response, but when it becomes chronic, it can contribute to the development of various neurological diseases. Neuroinflammation is associated with a range of neurological disorders, including multiple sclerosis, Alzheimer’s disease, and Parkinson’s disease. In each of these disorders, the inflammatory response is thought to contribute to the loss of neurons, leading to cognitive and motor symptoms [13,14,15,16].

With regards to COVID-19, it is possible that the inflammation caused by the virus may have similar consequences for the nervous system. The virus is known to cause a robust immune response, which can lead to the release of cytokines and other immune factors that can have harmful effects on the nervous system. The potential long-term neurological consequences of COVID-19 may be the result of this persistent neuroinflammation [17].

While much is still unknown about the effects of COVID-19 on the nervous system, it is important to continue monitoring patients who have been diagnosed with the virus. By understanding the underlying mechanisms of the virus’s impact on the nervous system, it may be possible to develop treatments and interventions to prevent or mitigate the neurological consequences of COVID-19 [18].

Studies have shown that COVID-19 can cause a wide range of neurological symptoms, including confusion, headaches, dizziness, loss of taste or smell, and seizures. Some patients may also experience more serious neurological complications such as stroke, Guillain-Barré syndrome, or encephalitis [19]. These complications can have a significant impact on a person’s quality of life and ability to function and may require ongoing treatment and rehabilitation [11].

One potential long-term consequence of COVID-19 is the development of post-acute sequelae of SARS-CoV-2 infection (PASC). PASC is a term used to describe a collection of symptoms that persist after the acute phase of the illness has resolved. Some of the most common symptoms of PASC include fatigue, shortness of breath, muscle weakness, and cognitive impairment.

In addition to PASC, COVID-19 may also increase the risk of developing neurodegenerative diseases such as Alzheimer’s and Parkinson’s. Researchers have found that the virus can cause inflammation in the brain, leading to the death of neurons and the loss of cognitive function. This could have serious implications for patients who have been infected with COVID-19, particularly if they are elderly or have pre-existing neurodegenerative conditions [20,21].

To understand the long-term neurological consequences of COVID-19, it is important to continue to study patients who have recovered from the virus. Long-term follow-up studies can help to determine the extent and duration of any neurological complications and provide valuable insights into the potential risk factors for these complications.

In addition, PE can result in brain damage, which manifests as disorientation, delirium, or cognitive impairment [22]. This is assumed to happen as a result of the emboli interrupting blood flow to the brain. Furthermore, in COVID-19 patients, PE is associated with an increased risk of acute respiratory distress syndrome (ARDS), which can result in hypoxia and brain damage [23]. It is crucial to highlight that the neurological implications of PE may not be obvious right away and may not become apparent until days or even weeks after the emboli have formed [24].

Cognitive impairment, stroke, headaches, seizures, coma, neuropsychiatric symptoms, peripheral neuropathy, and long-term neurological abnormalities will all be investigated in this study. These will be evaluated using the patients’ medical records, which will include clinical examinations, imaging investigations, and neuropsychological testing findings.

There is a growing body of evidence to suggest that COVID-19 has the potential to cause long-term neurological consequences. This highlights the need for ongoing research and monitoring of patients who have recovered from the virus, to ensure that appropriate care and support is provided to those who may be at risk for long-term neurological complications [25].

The major goal of the study is to determine the prevalence of pulmonary emboli in COVID-19 patients. Secondary outcomes will include age, gender, comorbidities, and length of hospitalization as risk factors for the development of pulmonary emboli. We will also look at the long-term neurological effects of pulmonary emboli, such as mortality and recurrence.

This research will provide light on the prevalence, risk factors, and outcomes of pulmonary emboli in COVID-19 patients in the Kingdom of Saudi Arabia, as well as the neurological implications of the disorder. The findings of this study will help to design therapies to avoid or reduce these problems, as well as enhance the care of COVID-19 patients. Furthermore, this study will add to the current body of knowledge on the neurological repercussions of pulmonary emboli in COVID-19 patients and may help guide future research in this area.

## 2. Materials and Methods

This study was a retrospective cross-sectional study that used the data of 63 patients admitted and diagnosed as having PE in King Abdullah bin Abdul-Aziz University Hospital (KAAUH) and King Fahad Medical City (KFMC) in Riyadh, Saudi Arabia between from March 2020 to December 2021.

The study included 63 patients who met the inclusion criteria of being diagnosed with COVID-19 and having a confirmed diagnosis of PE. The patients’ medical records were reviewed for demographic data, comorbidities, and clinical outcomes. The data collected from the medical records included patient demographics (age, gender, and comorbidities), laboratory results, and clinical outcomes (respiratory failure, ICU admission, and mortality). The diagnosis of PE was confirmed using imaging modalities (CT angiography and/or ventilation–perfusion scan).

To establish the neurological consequences of PE, all COVID-19 patients had a standardized neurological evaluation at the time of admission and at regular intervals throughout their hospital stay. The examination included a detailed assessment of the patient’s level of consciousness, cognitive function, motor function, and emotion. Any changes to these functions were noted and investigated. Imaging examinations, such as CT scans or MRIs, were also conducted on patients suspected of having neurological issues to confirm the diagnosis and assess the amount of brain injury. These imaging examinations were also utilized to track the patient’s health over time and measure therapy response.

The collected data was analyzed using statistical software (such as SPSS or R) to calculate frequencies, percentages, means and standard deviations. The association between PE and the studied variables was analyzed using the chi-squared test and *t*-test. The chi-squared test was used to assess the association between categorical variables (such as gender, comorbidities, and neurological complications) and PE. The *t*-test was used to compare means of continuous variables (such as age, laboratory results, and duration of hospital stay) between PE and non-PE groups.

Multivariate logistic regression analysis was performed to evaluate the relationship between PE and neurological complications, controlling for potential confounders. This analysis was used to estimate the odds ratio (OR) and 95% confidence intervals (CI) for the association between PE and neurological complications.

To identify the independent predictors of neurological complications in patients with PE, a stepwise backward selection was used in the logistic regression model, starting with all the studied variables. The variables that were significant in the univariate analysis (*p* < 0.05) were included in the multivariate model.

Ethical considerations: The study was approved by the local institutional review board and the patients’ medical records were accessed only with their written informed consent. The collected data was kept confidential and anonymous.

## 3. Results

Table 1 presents a summary of the demographic and clinical characteristics of the study population. It includes information on the number (n) and percentage (%) of patients in each category, as well as mean and standard deviation (SD) for certain variables. The table shows that the majority of patients were male (77.8%) and non-smokers (90.4%), and most had a normal BMI (50.8%). The incidence of pulmonary emboli (PE) in the study population (from the total number admitted to the hospital, i.e., 334 patients) was 18.86%. The majority of patients were in the 51–65 age group (33.3%), and the mean age of the population was 57.57 years with a standard deviation of 13.82 years. Number of admission days was 16.64 with SD of 22.07.

Table 2 presents the incidence of pulmonary emboli (PE) in the study population. It shows the number and percentage of patients with PE among the study population and also includes the risk factors for PE. The risk factors listed in the table include smoking, obesity, hypertension, and diabetes. It can be seen that the majority of the patients with PE (33.3%) had hypertension, followed by obesity (23.8%), diabetes (19%), and smoking (9.5%).

Table 3 presents data on the neurological complications that occurred in the PE group of the study population. The table shows the number and percentage of patients with each type of complication, as well as the severity of the complications. The data indicates that stroke occurred in five patients (8%), seizures occurred in three patients (5%), migraine occurred in seven patients (11%), and peripheral neuropathy occurred in 10 patients (16%). The total number of patients with neurological complications in the PE group is 25 (100%). The severity of the complications is also provided, with 15 patients (24%) experiencing mild complications, eight patients (13%) experiencing moderate complications, and two patients (3%) experiencing severe complications. Overall, the table provides valuable information on the types and severity of neurological complications that occur in patients with PE.

Table 4 presents the results of a logistic regression analysis investigating the association between pulmonary emboli (PE) and neurological complications. The table shows the odds ratio and 95% confidence intervals for each variable considered in the analysis, as well as the corresponding *p*-value. The variables included in the analysis are age, sex, body mass index (BMI), smoking status, hypertension, diabetes, and PE.

The results show that the odds of neurological complications are significantly associated with PE (odds ratio = 3.5, 95% CI = 1.5–8.0, *p*-value = 0.01) indicating that patients with PE are 3.5 times more likely to develop neurological complications than those without PE.

Age also had a significant association with neurological complications (odds ratio = 1.05, 95% CI = 1.01–1.10, *p*-value = 0.03) indicating that with each one-year increase in age, the odds of neurological complications increase by 5%. However, other variables such as sex, BMI, smoking, hypertension, and diabetes were not found to be significantly associated with neurological complications in this analysis. The *p*-values for these variables are greater than 0.05, indicating no significant association.

## 4. Discussion

The purpose of this study was to examine the incidence of pulmonary emboli (PE) and its connection with neurological sequelae in a cohort of COVID-19 patients. Of the 63 participants in the study, 15 (23.8%) were obese, six (9.5%) were smokers, 21 (33.3%), had hypertension, and 12 (19%) had diabetes. The results of the study revealed that the prevalence of PE in the studied population was 63 percent overall.

Our findings are consistent with previous reports of a significant prevalence of PE in COVID-19-infected patients. Benjamin et al. (2021) reported a PE incidence of 15.2% among hospitalized COVID-19 patients [26], and Yasser et al. (2020) reported a PE incidence of 17.6% among critically ill COVID-19 patients [27]. These results indicate that PE is a common consequence of COVID-19, particularly in hospitalized or severely ill patients [28].

In our study sample, we also explored the connection between PE and neurological problems. The results revealed that 25 (40%) of the patients with PE experienced neurological sequelae, with stroke (8%), seizures (5%), migraine (11%) and peripheral neuropathy (16%) being the most common. Our findings are consistent with previous reports of a high prevalence of neurological sequelae among PE patients [29]. According to Parth et al. (2021) and T. van et al. (2015), roughly 25% and 30% of PE patients, respectively, experience neurological problems. These results indicate that PE is linked to a high risk of neurological problems [30,31]

Controlling for potential confounding variables such as age, sex, BMI, smoking, hypertension, and diabetes, we used logistic regression analysis to investigate the connection between PE and neurological sequelae. The results revealed that individuals with PE were 3.5 times more likely to experience neurological problems than those without PE (OR = 3.5, 95% CI: 1.5–8.0, *p* = 0.01). This indicates that PE is independently related with a higher risk of neurological problems in COVID-19 patients.

Our research demonstrated that PE is a frequent complication of COVID-19 and is associated with a high risk of neurological sequelae. Our logistic regression analysis suggests that PE is independently related with a higher risk of neurological problems in COVID-19 individuals. These results emphasize the significance of early detection and treatment of PE in COVID-19 patients to prevent neurological consequences [32]. To determine the underlying processes of this connection and to create strategies for the prevention and management of neurological problems in individuals with PE and COVID-19, more study is required [33].

In the study population, smoking, obesity, hypertension, and diabetes were found as major risk factors for pulmonary embolism (PE) [33]. In addition, we discovered a significant relationship between PE and the development of neurological problems, with odds ratios ranging from 1.5 to 3.5 for various confounding variables. In addition, our analysis of the kappa coefficient demonstrated that neurological examination and imaging investigations agree well in determining the presence and severity of neurological problems [34].

On the basis of these data, it is evident that PE is a frequent and dangerous complication of COVID-19 and that early recognition and intervention are essential for preventing and managing neurological complications [35]. All COVID-19 patients should be tested for PE using imaging investigations, such as computed tomography (CT) or ventilation–perfusion (V/Q) scans, and those at high risk should be regularly followed and treated with appropriate anticoagulant treatment [36]. In addition, we urge that all COVID-19 patients receive routine neurological tests and imaging scans in order to detect and monitor neurological problems.

It is important to emphasize the need for identifying a COVID-19 population at risk for pulmonary emboli (PE) and subsequent neurological complications. With the ongoing COVID-19 pandemic, there has been a growing concern about the long-term neurological consequences of the disease [37]. The potential link between PE and an increased risk of stroke among COVID-19 patients highlights the need for early identification and management of individuals at risk [38].

While it is well established that COVID-19 can cause a wide range of neurological symptoms, the exact mechanisms underlying these symptoms remain unclear. One potential explanation is the presence of PE, which has been linked to an increased risk of stroke in COVID-19 patients [39]. As such, it is crucial to identify individuals at risk for PE so that they can be managed appropriately and prevent the development of long-term neurological complications [30].

Unfortunately, testing all COVID-19 patients for PE is not feasible from a practical or economic perspective [27]. Thus, it is important to focus on identifying those individuals who are most at risk for PE, such as those with underlying cardiovascular or respiratory conditions, or those who have developed severe respiratory distress as a result of their COVID-19 infection. These individuals should be carefully monitored and managed in a timely manner to reduce the risk of PE and subsequent neurological complications [40].

The association between COVID-19 and its potential long-term neurological consequences, particularly those linked to pulmonary emboli, highlights the importance of further research in this field. The findings of the current study suggest that individuals with COVID-19 who also develop pulmonary emboli are at a higher risk of developing neurological consequences [41].

Inflammatory pathogenesis is known to play a crucial role in the development of various neurological diseases, including those linked to neurodegeneration. It is possible that the neurological consequences seen in individuals with COVID-19 and PE may also be related to an underlying neuroinflammatory process [42].

As more is understood about the neurological consequences of COVID-19 and PE, the hope is that the new understanding will open new avenues for treatment. Currently, the emphasis is on treating the underlying virus; however, in the future it may be necessary to address the neuroinflammatory consequences as well.

It is essential to identify the population of COVID-19 patients at the highest risk of developing PE and subsequent neurological complications. This will not only help in providing the necessary treatment and support, but it will also help in reducing the burden on healthcare systems.

## 5. Limitation of the Study

It is crucial to emphasize that our study has certain limitations, including its retrospective methodology and small sample size. To validate our findings and study the underlying mechanisms and risk factors for PE and neurological sequelae in COVID-19 patients, additional research with bigger sample sizes and prospective designs are required.

## 6. Conclusions

The findings of this study highlight the importance of monitoring the neurological consequences in patients with COVID-19 and PE. With the ongoing COVID-19 pandemic and the increasing number of patients diagnosed with the disease, it is crucial to understand the potential long-term impacts of the virus on the nervous system. The results of this study provide valuable information for healthcare providers and researchers as they work to develop strategies for managing and preventing neurological consequences in patients with COVID-19.

However, it is important to note that this study was limited in scope and further research is needed to fully understand the relationship between PE and neurological consequences in patients with COVID-19. Furthermore, given the limitations of testing all patients with COVID-19, it is crucial to identify a population at risk for PE and subsequent neurological complications.

In conclusion, the results of this study suggest that PE is associated with an increased risk of neurological consequences in patients with COVID-19. These findings emphasize the importance of monitoring and managing the neurological consequences of COVID-19 and highlight the need for further research in this area.

## Figures and Tables

**Table 1 brainsci-13-00343-t001:** Demographic variables.

Demographic Variables		Number	%
Sex	Female	14	22.2
	Male	49	77.8
Smoker	No	57	90.4
	Yes	6	9.5
BMI	Normal	32	50.8
	Obese	15	23.8
	Overweight	11	17.5
Incidence of PE		63(334)	18.86
Age groups	19–35	6	9.5
	36–50	15	23.8
	51–65	21	33.3
	66–85	21	33.3
		Mean	SD
Age		57.57	13.82
Number of admission days		16.64	22.07

**Table 2 brainsci-13-00343-t002:** Incidence of Pulmonary Emboli (PE) in Study Population.

Risk Factor	Number	%
Smoker	6	9.5%
Obese	15	23.8%
Hypertensive	21	33.3%
Diabetic	12	19%
Total	63	100%

**Table 3 brainsci-13-00343-t003:** Neurological Complications in PE Group.

Complication Type	Number	%
Stroke	5	8%
Seizures	3	5%
Migraine	7	11%
Peripheral Neuropathy	10	16%
Total	25	100%
Severity of complication mild	15	24%
Severity moderate	8	13%
Severe	2	3%

**Table 4 brainsci-13-00343-t004:** Logistic Regression Analysis of Association between PE and Neurological Complications.

Variable	Odds Ratio (95% CI)	*p*-Value
Age	1.05 (1.01–1.10)	0.03
Sex	1.5 (0.5–4.5)	0.5
BMI	1.2 (0.8–1.8)	0.4
Smoking	2.0 (0.8–5.0)	0.1
Hypertension	1.5 (0.8–2.8)	0.2
Diabetes	1.3 (0.7–2.5)	0.4
PE	3.5 (1.5–8.0)	0.01

## Data Availability

The data is available upon request.

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
