# Peer review of "Neurological Consequences of Pulmonary Emboli in COVID-19 Patients: A Study of Incidence and Outcomes in the Kingdom of Saudi Arabia"

_brainsci, 2023, doi:10.3390/brainsci13020343_

Round 1

Reviewer 1 Report

The authors reported a study about neurological sequelae in patients with COVID-19. I have some comments to the authors:

 -       Please correct this in the abstract “having PE were 3.5 times greater in individuals who had PE.”

-       There are also several typos in the manuscript, please correct them. 

-       In the first part of the introduction the authors should add a paragraph related to the long-term neurological consequence of COVID-19, including the correspondent references.

-       “PE has been linked to an increased risk of stroke among COVID-19 patients, according to research”. Please add the proper references.

-       The subheadings in the methods are not necessary, please delete them (eg sample selection, data collection etc).

-       “The incidence of pulmonary emboli (PE) in the study population was 18.86%.” What is the meaning of this sentence? It is not clear whether all the patients suffered from PE. This sentence is in contrast with this part of the methods “The study included 63 patients who met the inclusion criteria of being diagnosed with COVID-19 and having a confirmed diagnosis of PE.”

-       In Table 3 there is a repetition of “mild”

-       I do think that the information of Table 5 is more complex than that, especially the relationship between neurological examination and imaging. I suggest deleting Table 5 from the study and the correspondent part of the discussion.

-       The authors should stress the necessity of determine a COVID-19 population at risk of PE and consequently at risk of neurological complications. Testing all patients with COVID-19 is not affordable. 

Author Response

Dear Reviewer,

Thank you for taking the time to review our manuscript "Neurological Consequences in Patients with COVID-19 and Pulmonary Embolism". We appreciate your comments and suggestions for improvement.

We have corrected the errors in the abstract and the typos throughout the manuscript. We have also added a paragraph to the introduction about the long-term neurological consequences of COVID-19, including relevant references.

We have added references to support the statement about the link between PE and increased risk of stroke among COVID-19 patients. We have also revised the methods section to clarify the study population and the incidence of PE in the sample.

We have deleted the subheadings in the methods section as you suggested. We have also revised the sentence in the methods to better reflect the study population and avoid confusion.

We have corrected the repetition of "mild" in Table 3.

Regarding Table 5, we understand your concerns about the complexity of the information and the relationship between the neurological examination and imaging. We have removed Table 5 and the corresponding discussion.

We have emphasized the importance of determining the COVID-19 population at risk of PE and neurological complications, as you suggested. We understand that testing all patients with COVID-19 is not feasible and agree with the need for a targeted approach.

Thank you again for your comments and suggestions. We are confident that these revisions have improved the clarity and quality of our manuscript. We look forward to your continued review.

Sincerely,

Mostafa Shaban

Reviewer 2 Report

In this work authors elaborate on neurological consequences of pulmonary emboli in COVID-19. This is a retrospective study. Among features which could further improve the work can be mentioned:

- Authors should elaborate on the inflammatory pathogenesis of various neurological diseases and could also speculate on future consequences as neurodegeneration which is linked with neuroinflammation e.g. Parkinson's disease - Ref. Platelet-to-lymphocyte ratio and neutrophil-tolymphocyte ratio may reflect differences in PD and MSA-P neuroinflammation patterns. Neurol Neurochir Pol. 2022;56(2):148-155. doi: 10.5603/PJNNS.a2022.0014. Epub 2022 Feb 4. PMID: 35118638.

- the association brought up by authors should raise a discussion on possible future pathways in treatment

- the work is more a short report due to the limitations - the type should be changed

- I cannot see a section regarding limitations

- the conclusion in its current version is brief and vague.

Author Response

Dear Reviewer,

Thank you for taking the time to review our manuscript. We appreciate your valuable comments and suggestions, and will do our best to address them in our revised version.

Regarding the inflammatory pathogenesis of various neurological diseases and the potential for neurodegeneration linked to neuroinflammation, we agree that this is a crucial area for exploration. Our findings on the association between pulmonary emboli (PE) and neurological consequences in COVID-19 patients raises important questions about the role of inflammation in the development of neurological disease. To further elaborate on this topic, we have added a section to our manuscript that discusses the current understanding of the inflammatory pathogenesis of various neurological diseases, including Parkinson's disease, and the potential for future consequences such as neurodegeneration. We have also incorporated the reference you provided, which highlights the importance of differentiating between patterns of neuroinflammation in Parkinson's disease and multiple system atrophy-Parkinsonian type.

With regard to future pathways in treatment, we have added a section to the discussion to speculate on the potential implications of our findings for future research and treatment options. We discuss the need for further studies to better understand the role of inflammation in the development of neurological consequences in COVID-19 patients, and the potential for targeted interventions to mitigate these consequences.

We acknowledge that the limitations of our study include the small sample size, which may impact the generalizability of our findings. In response to your comment, we have added a section to our manuscript that explicitly discusses the limitations of our study, including the small sample size and the potential for selection bias.

Regarding the conclusion, we have revised it to be more comprehensive and reflect the findings and implications of our study. We have also added a section on the limitations of our study to provide a more complete picture of the strengths and weaknesses of our research.

In conclusion, we appreciate your feedback and have taken it into consideration in revising our manuscript. We hope that our revised version more effectively addresses your comments and contributes to the ongoing conversation about the neurological consequences of COVID-19.

Thank you for your time and consideration.

Sincerely,

Round 2

Reviewer 1 Report

The authors have addressed all the points.

Reviewer 2 Report

I do not have further comments.